# Quality performance and associated factors in Swiss diabetes care – A cross-sectional study

**Rahel Meier**[ID]*, **Fabio Valeri, Oliver Senn, Thomas Rosemann, Corinne Chmiel**[ID]

Institute of Primary Care, University of Zurich and University Hospital Zurich, Zürich, Switzerland

* rahel.meier@usz.ch

## Abstract

### Introduction

Quality indicators and pay-for-performance schemes aim to improve processes and outcomes in clinical practice. However, general practitioner and patient characteristics influence quality indicator performance. In Switzerland, no data on the pay-for-performance approach exists and the use of quality indicators has been marginal. The aim of this study was to describe quality indicator performance in diabetes care in Swiss primary care and to analyze associations of practice, general practitioner and patient covariates with quality indicator performance.

### Methods

For this cross-sectional study, we used medical routine data from an electronic medical record database. Data from 71 general practitioners and all their patients with diabetes were included. Starting in July 2018, we retrieved 12-month retrospective data about practice, general practitioner and patient characteristics, laboratory values, comorbidities and co-medication. Based on this data, we assessed quality indicator performance of process and intermediate outcomes for glycated hemoglobin, blood pressure, cholesterol and associations of practice, general practitioner and patient characteristics with individual and cumulative quality indicator performance. We calculated odds ratios (OR) and 95% confidence intervals (CI) using regression methods.

### Results

We assessed 3,383 patients with diabetes (57% male, mean age 68.3 years). On average, patients fulfilled 3.56 (standard deviation: 1.89) quality indicators, whereas 17.2% of the patients fulfilled all six quality indicators. On practice and general practitioner level, we found no associations with cumulative quality indicator performance. On patient level, gender (ref = male) (OR: 0.83, CI: 0.78–0.88), number of treating general practitioners (OR: 0.94, CI: 0.91–0.97), number of comorbidities (OR: 1.43, CI: 1.38–1.47) and number of consultations (OR: 1.02, CI: 1.02–1.02) were associated with cumulative quality indicator performance.

**Data Availability Statement:** The data was gathered within the ongoing FIRE project. The FIRE database can be accessed at any time by the scientific team of the institute. Legal restrictions in Switzerland prohibit public release of original

patient data without consent. The authors' fully anonymized data is exempt from these legal restrictions. However, the data could be deanonymized by individuals or organizations, such as health insurers which have overlapping data (e.g. patient date of birth and consultation dates). Data access queries can be addressed to Rahel Meier (rahel.meier@usz.ch) after clearance by the local ethics committee or to the Kantonale Ethikkommission Zurich (Local Ethics Committee of the Canton of Zurich) (Info.KEK@kek.zh.ch).

**Funding:** This project is supported by a grant from the national research program 'Smarter health care' from the Swiss national science foundation (SNSF, www.snsf.ch), grant number 407440_167204. TR received the grant. The funders had no role in study design, data collection and analysis, decision to publish, or preparation of the manuscript.

**Competing interests:** The authors have declared that no competing interests exist.

## Conclusion

The influence of practice, general practitioner and patient characteristics on quality indicator performance was surprisingly small and room for improvement in quality indicator performance of Swiss general practitioners seems to exist in diabetes care.

## Introduction

The use of quality indicators (QI) and contingent incentives aim to improve processes and outcomes in clinical practice. However, whether QI performance is modifiable by introducing a pay-for performance (P4P) scheme is still unclear [1–4]. Previous studies showed that P4P programs' effectiveness highly depend on type of health care system, investigated QI, study participants, patient population and the level of payment [2, 5, 6]. A systematic review [7] concluded that financial incentives targeting process and intermediate outcome indicators yield the highest effect, as they can be directly influenced by general practitioners (GP). In diabetes, which is one of the most common diseases for assessing quality of care, the most frequent process and intermediate outcome QIs are for glycated hemoglobin (HbA1c), blood pressure (BP) and serum cholesterol levels. From literature, we know that not only practice and GP, but also patient characteristics influence QI performance [8–12]. Case-mix adjustments are therefore often used to control for these mechanisms. However, the case-mix adjustments used and their effect on QI performance vary widely across studies [13–15].

In Switzerland, the use of QIs, especially in primary care, has been marginal. No P4P approach exists and no case-mix adjustments have been investigated [16]. Currently, a randomized controlled trial (RCT) testing the P4P approach in Swiss primary care using clinical routine data is ongoing [17]. The baseline data of this trial offer the opportunity to study the characteristics and QI performance of the study population and to analyze associations of practice, GP and patient covariates with QI performance.

## Methods

### Study design and setting

For this cross-sectional study we used baseline data collected within a cluster randomized controlled trial (trial registration number: ISRCTN13305645) [17]. Unit of cluster randomization was at practice level. The baseline assessment covered 12 months retrospectively (Fig 1). The trial was launched within the family medicine international classification of primary care (ICPC) research using electronic medical records (EMR) (FIRE) database of the Institute of Primary Care of the University of Zurich [18]. Up to June 2018, more than 400 GPs from the German-speaking area of Switzerland participated in the FIRE project. The participants contribute anonymized data containing the following components: administrative information, vital signs, laboratory values, medication data and diagnostic codes coded according to ICPC-2 [19]. Up to June 2018, information from more than 500,000 patients and 5 million consultations are available. Further, at individual project entry the participating GPs manually provided additional information concerning structural details to the study nurse of the research team. According to the local ethics committee of the canton of Zurich, the project does not fall under the scope of the law on human research and therefore no ethical consent was necessary (BASEC-Nr. Req-2017-00797).

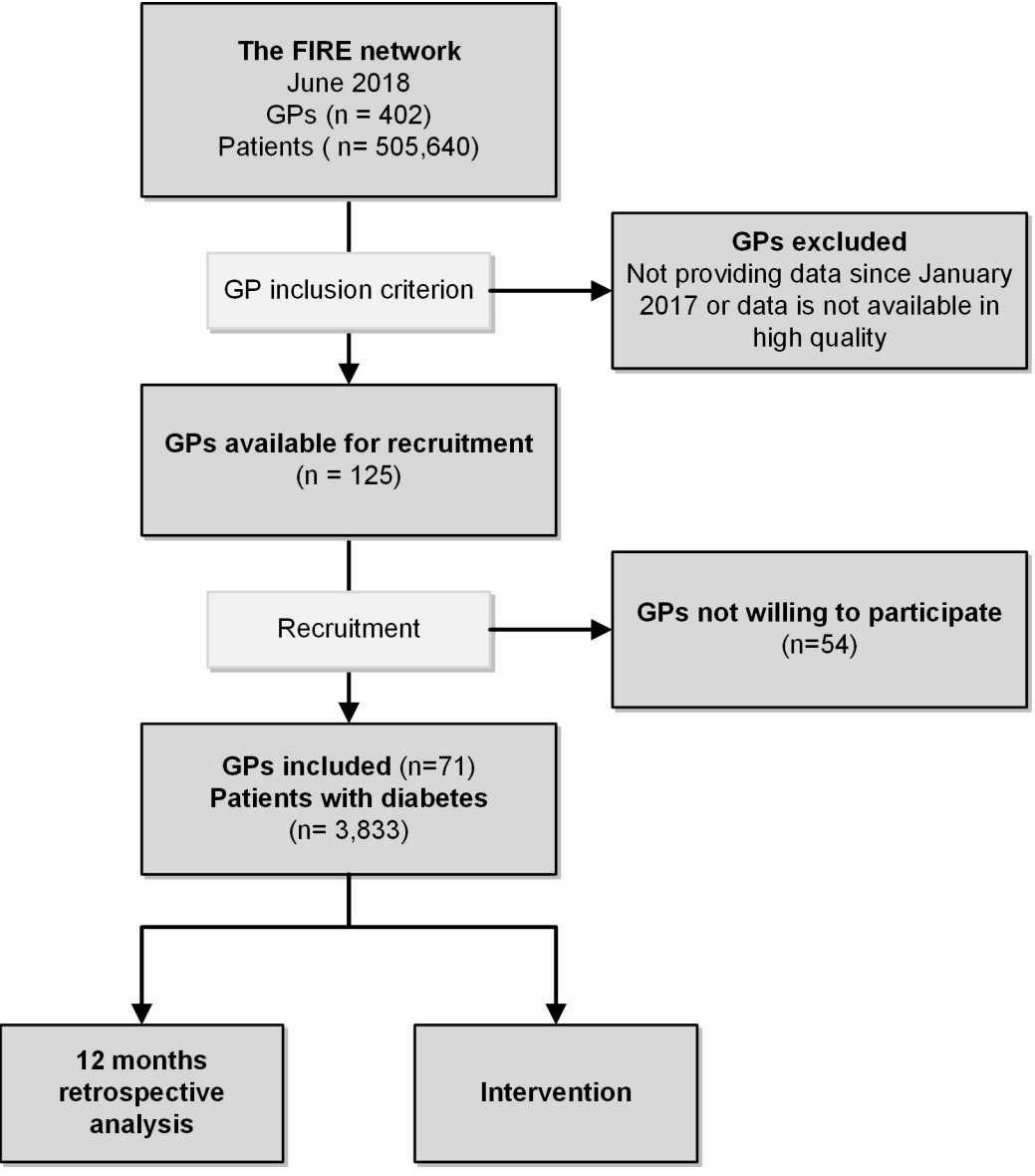

**Fig 1. Flowchart.** Study design including inclusion/exclusion criteria, FIRE = Family medicine ICPC research using electronic medical records; GP = General practitioner.

## Participants

For the trial, the following data availability and data quality criteria had to be fulfilled on GP level: a) continuous data delivery since January 2017, b) delivering HbA1c and BP values in more than 10% of their patients with diabetes, and c) consulting a minimum of five patients with diabetes. In June 2018, the eligible GPs received an invitation to participate in the study. Per practice, multiple GPs were contacted if data availability and data quality criteria were fulfilled (Fig 1). From the participating GPs, all patients diagnosed with diabetes at least 4 months before the baseline date were subsequently identified according to at least one of the following criteria:

1. Patients with ICPC-2 codes T89 (insulin dependent diabetes mellitus) and T90 (insulin independent diabetes mellitus)

2. Patients with antidiabetic medication (oral antidiabetics and/or insulin) according to the anatomical therapeutic chemical (ATC) classification system (A10A, A10B, A10X) [20]

## Database query and variables

From the included GPs, we retrieved demographic (year of birth, gender) and work setting related data (practice location to assess urbanity [21], practice type (single or group practice), physician's network participation). From the included patients we retrieved following data: a) demographics (year of birth, gender), b) laboratory values(HbA1c, cholesterol) and vital signs (BP, body mass index (BMI) recorded within the observation period, c) prescription of relevant medication (insulin, oral anti-diabetic medication, anti-hypertensive medication, anti-thrombotic medication, lipid lowering medication) recorded within the observation period, d) presence of comorbidities relevant for diabetes (obesity, arterial hypertension, hyperlipidemia, microvascular complications (chronic kidney disease stage 3a or higher, peripheral vascular disease, retinopathy or neuropathy) and macrovascular complications (coronary heart disease (CHD), chronic heart failure (CHF), stroke) available in the full patient history in the FIRE database. Detailed information about the identification scheme for comorbidities is depicted in the S1 Table. For each patient, we determined the baseline performance (fulfilled/not fulfilled) based on the QI defined in the P4P trial [22, 23]. QIs are listed in Table 1. Furthermore, we calculated the cumulative QI performance, which is the number of fulfilled QIs per patient.

## Objectives

Objectives of the current study are:

• Description of the study population characteristics, including baseline QI performance

• Examination of the associations of practice, GP and patient covariates with QI performance

## Statistical analysis

We presented categorical data as frequencies and percentages, continuous variables as means and standard deviations (SD) or median and interquartile range (IQR), as appropriate. Overall QI performances are expressed as percentage of patients meeting the indicator (numerator), in reference to all eligible patients (denominator). We used hierarchical multivariable logistic regression models, with the practice and the GP nested within practices as random variables, to examine the independent association of each QI performance on patient level with practice (practice location, practice type), GP (age, sex, physician's network participation) and patient characteristics (age, sex, number of comorbidities, number of consultations, number of consulted GPs). Number of medications and BMI were not considered as covariates for the regression model, since they were

**Table 1. Quality indicators used to assess performance.**

|  | Process indicators | Outcome indicators |
|---|---|---|
| Blood pressure | Proportion of patients with diabetes with at least one blood pressure measure-meant in the preceding 12 months. | Proportion of patients with diabetes with a blood pressure measurement < 140/85 mmHg in the preceding 12 months. |
| HbA1c | Proportion of patients with diabetes with at least one measurement of HbA1c in the preceding 12 months. | Proportion of patients with diabetes with HbA1c levels < 7.5% in the preceding 12 months. |
| Cholesterol | Proportion of patients with diabetes with at least one cholesterol measurement in the preceding 12 months. | Proportion of patients with diabetes with total cholesterol < 5 mmol/l in the preceding 12 months. |

HbA1c = measure for glycated hemoglobin;

used to identify certain comorbidities. We ran the model for each of the six QIs. The same model variables were used in a hierarchical multivariable binomial logistic regression model to assess the association for the cumulative QI performance, whereas the dependent variable was defined as the cumulative QI performance. We reported odds ratios (OR) and 95% confidence intervals (CI) for each factor included in the model and used random effects to study variance on practice and GP level. All analyses were performed using the statistical software R (version 3.5.0) [24].

## Results

### Sample characteristics

We included 71 GPs from 43 practices in the study. We enrolled 61 GPs in cohort 1, starting in July 2018, whereas 10 GPs were enrolled in a second cohort, starting in September 2018. Their mean age was 52 years (SD 9.3), 72% were male and 92% worked in a group practice. They practiced in 83% in an urban setting and 93% were member of a physician's network. With the participating GPs, a total of 3,833 patients with diabetes were included. The median number of patients with diabetes per GP was 44 (IQR: 28–79), corresponding to 5% (IQR: 3% - 7%) of GP's patient list size. These patients were 57% male and had a mean age of 68.3 years (SD 13.4). The first record of diabetes within the database was on average 2.6 years (IQR: 1.3– 6.2) before baseline assessment.

93.8% of patients had at least one, 37.9% three or more comorbidities. The most frequent comorbidity was arterial hypertension, followed by hyperlipidemia and obesity (see Table 2 for exact numbers on comorbidities). Diabetes-associated microvascular complications were identified in 18.3% of patients, macrovascular complications in 10.0% (see Table 2 for exact numbers).

### Treatment and disease characteristics

On average, patients had eight consultations (IQR: 5–15) at the GPs' practice in the 12 months preceding baseline. In those consultations, average numbers of BP measurements, HbA1c testing, cholesterol testing and BMI measurements, as well as the parametric values thereof are reported in Table 2. Regarding anti-diabetic therapy, we found that 72.7% of patients received a therapy; most often oral medication only, followed by the combination of an oral and an insulin therapy and insulin only (see Table 2 for the exact numbers). Moreover, 47.9% of patients received an antihypertensive agent, 40.2% antiplatelet therapy and anticoagulants, and 35.5% lipid-lowering therapy. For proportions of patients achieving the defined QIs see Table 3. On average, patients fulfilled 3.6 (SD: 1.9) QIs, whereas 17.2% of the patients fulfilled all QIs.

### Associations with QI performance

The regression model revealed the following results: for practice and GP characteristics, we did not find evidence of significant associations with QI performance (Table 4), except for female GPs measuring BP more often (OR 1.75 95% CI 1.03–2.98) and older GPs achieving BP target level in a larger share of their patients (OR 1.23 95% CI 1.02–1.49). On patient level, age had no influence on achieving BP QIs (process indicator: OR 1.06 95% CI 0.99–1.13, outcome indicator: OR 1.00 95% CI 0.95–1.06). Higher age was significantly associated with achieving the HbA1c QIs more often (process indicator: OR 1.07 95% CI 1.01–1.14, outcome indicator: OR 1.10 95% CI 1.04–1.16), but with achieving the cholesterol QIs less often (process indicator: OR 0.86 95% CI 0.81–0.91, outcome indicator: OR 0.92 95% CI 0.86–0.97). Female gender was also significantly associated with achieving the cholesterol QIs more often (process indicator: OR 0.73 95% CI 0.63–0.85, outcome indicator: OR 0.5 95% CI 0.42–0.58). For patients

**Table 2. Patient, treatment and disease characteristics.**

| | Median, mean or n | IQR, SD or % |
|---|---|---|
| **Patient characteristics** | | |
| Male gender | 2198 | 57.3 |
| Age at baseline (years) | 68.3 | 13.4 |
| First record of diabetes before baseline (years) | 2.6 | 1.3–6.2 |
| **Diabetes associated Comorbidities** | | |
| Arterial hypertension | 3322 | 86.7 |
| Hyperlipidemia | 2262 | 59.0 |
| Obesity | 1589 | 41.5 |
| Chronic kidney disease | 454 | 11.8 |
| Peripheral vascular disease | 134 | 3.5 |
| Neuropathy | 100 | 2.6 |
| Retinopathy | 14 | 0.4 |
| Coronary heart disease | 234 | 6.1 |
| Heart failure | 83 | 2.2 |
| Stroke | 67 | 1.7 |
| **Treatment and disease characteristics** | | |
| Number of consultations | 8 | 5–15 |
| Number of BP measurements | 2.3 | 2.6 |
| Number of HbA1c measurements | 2.1 | 1.5 |
| Number of cholesterol measurements | 0.7 | 0.9 |
| Number of BMI measurements | 1.2 | 1.6 |
| Systolic BP value (mmHg) | 135.9 | 126.5–146.1 |
| Diastolic BP value (mmHg) | 80 | 73.7–85.0 |
| HbA1c value (%) | 6.8 | 6.3–7.5 |
| Cholesterol value (mmol/l) | 4.5 | 3.8–5.4 |
| BMI value (kg/m2) | 29.53 | 26.3–33.3 |
| **Diabetes associated medication** | | |
| Oral anti-diabetic medication | 1978 | 51.6 |
| Insulin | 278 | 7.5 |
| Combination of oral medication and insulin | 520 | 13.6 |
| Antihypertensive medication | 1837 | 47.9 |
| Antiplatelet therapy and anticoagulants | 1542 | 40.2 |
| Lipid lowering medication | 1359 | 35.5 |

IQR = interquartile range; SD = standard deviation; BP = blood pressure; HbA1c = measure for glycated hemoglobin; BMI = body mass index;

with an increasing number of diabetes-relevant comorbidities, the QIs were more often fulfilled (see Table 4 for OR & 95% CI), whereas the number of consultations only had a positive effect on fulfilling the BP and HbA1c QIs (Table 4). With an increasing number of GPs providing care for the same diabetes patient, the chances in achieving process indicators decreased (Table 4). Number of years since diabetes diagnosis was significantly associated with achieving HbA1c outcome QI less often (OR 0.96 95% CI 0.93–0.99), but achieving cholesterol outcome QI more often (OR 1.08 95% CI 1.04–1.12). Associations of the cumulative QIs are presented in Fig 2.

Hierarchical random effects for GPs nested in practices showed, that the variation on practice and GP level are considerable but different for each QI (see S1–S6 Figs). For the model

**Table 3. Proportion of patients achieving the defined quality indicators.**

| Description | Quality indicator performance % (n) |
|---|---|
| Proportion of patients with diabetes with at least one blood pressure measurement in the preceding 12 months. | 75.6 (2,899) |
| Proportion of patients with diabetes with a blood pressure measurement < 140/85 mmHg in the preceding 12 months. | 50.6 (1,941) |
| Proportion of patients with diabetes with at least one measurement of HbA1c in the preceding 12 months. | 80.4 (3,082) |
| Proportion of patients with diabetes with HbA1c levels < 7.5% in the preceding 12 months. | 66.3 (2,543) |
| Proportion of patients with diabetes with at least one cholesterol measurement in the preceding 12 months. | 49.3 (1,891) |
| Proportion of patients with diabetes with total cholesterol < 5 mmol/l in the preceding 12 months. | 33.5 (1,285) |

HbA1c = measure for glycated hemoglobin;

with the cumulative QI, only little variation was associated with the practice level and the unexplained variation was associated with the GP level (see S7 Fig).

## Discussion

In this study, we explored associations of practice, GP and patient characteristics with QI performance. We found no substantial effect from GP and practice characteristics on QI performance, whereas several patient characteristics had a small effect.

The patient population included in our study is highly comparable to the patient population of a recent RCT and a cross-sectional study in Swiss primary care [25, 26] in terms of age, gender, comorbidities and consultation count. In terms of age and gender, our study population was also highly comparable to other European diabetes populations, whereas in terms of comorbidities our population had less micro- and macro-vascular diseases [27]. Number of years since diabetes diagnosis was much shorter in our study, which is explained by the fact limited data were available from before GPs participated in the FIRE project.

Comparing QI performances from our study with previous studies is fairly challenging, due to heterogeneity regarding study type, patient population, clinical thresholds and underlying financial incentives of different health care systems [13, 27, 28]. The proportions of patients fulfilling the process and outcome indicators for BP and HbA1c were highly comparable to the methodological similar study of van Doorn-Klomberg et al. 2015 [13], whereas the European cross-sectional study of Stone et al. 2013 [27] reported process indicators above 90%. A recent Swiss study based on insurance claims data found slightly higher annual rates for HbA1c and total cholesterol measurements [29]. When comparing to the Swiss quality and outcome feasibility study of Djalali et al. 2014, which based on the same EMR database, an improvement for each QI was achieved [16]. However, QI performance in our study still showed room for improvement, especially for outcome indicators. Possible reasons for such poor performances might be clinical inertia to intensify treatment, poor patient adherence, or failure in structural data capturing [16, 30, 31].

Regression analysis revealed that included characteristics had no or very little effect on process and outcome indicators. Most significant effects were found on patient level, and the greatest positive effect on QI performance was an increasing number of diabetes-relevant comorbidities. More intensive consultations or an increased awareness and risk factor management in multimorbid patients with certain comorbidities might explain this finding [32,

**Table 4. Results of hierarchical multivariable regression analysis of quality indicator performance.**

| | OR | 95% CI | P-value | OR | 95% CI | P-value |
|---|---|---|---|---|---|---|
| | Blood pressure | | | | | |
| | Process indicator | | | Outcome indicator | | |
| Practice type (ref = group practice) | 2.07 | 0.86–4.97 | 0.10 | 1.41 | 0.75–2.64 | 0.29 |
| Practice location (ref = urban) | 0.90 | 0.47–1.74 | 0.76 | 0.83 | 0.51–1.35 | 0.45 |
| Network participation (ref = yes) | 0.74 | 0.30–1.84 | 0.52 | 1.09 | 0.55–2.17 | 0.81 |
| GP gender (ref = male) | 1.75 | 1.03–2.98 | <0.05 | 1.38 | 0.93–2.05 | 0.11 |
| GP age (per 10 years) | 1.11 | 0.86–1.43 | 0.43 | 1.23 | 1.02–1.49 | <0.05 |
| Patient gender (ref = male) | 0.90 | 0.76–1.07 | 0.25 | 1.00 | 0.87–1.15 | 0.98 |
| Patient age (per 10 years) | 1.06 | 0.99–1.13 | 0.09 | 1.00 | 0.95–1.06 | 0.93 |
| Number of GPs | 0.89 | 0.82–0.96 | <0.01 | 0.98 | 0.92–1.05 | 0.58 |
| Number of consultations | 1.05 | 1.04–1.07 | <0.001 | 1.06 | 1.05–1.07 | <0.001 |
| Number of comorbidities | 1.76 | 1.61–1.93 | <0.001 | 1.15 | 1.07–1.23 | <0.001 |
| Number of years since diagnosis | 0.98 | 0.94–1.01 | 0.21 | 1.02 | 0.98–1.05 | 0.32 |
| | HbA1c | | | | | |
| | Process indicator | | | Outcome indicator | | |
| Practice type (ref = group practice) | 0.71 | 0.39–1.31 | 0.28 | 0.75 | 0.46–1.24 | 0.26 |
| Practice location (ref = urban) | 0.75 | 0.45–1.23 | 0.26 | 1.05 | 0.71–1.56 | 0.81 |
| Network participation (ref = yes) | 0.94 | 0.48–1.82 | 0.85 | 0.78 | 0.45–1.35 | 0.38 |
| GP gender (ref = male) | 1.39 | 0.94–2.05 | 0.10 | 1.32 | 0.93–1.87 | 0.12 |
| GP age (per 10 years) | 0.90 | 0.74–1.08 | 0.24 | 0.95 | 0.81–1.11 | 0.51 |
| Patient gender (ref = male) | 0.85 | 0.72–1.02 | 0.08 | 0.91 | 0.79–1.05 | 0.19 |
| Patient age (per 10 years) | 1.07 | 1.01–1.14 | <0.05 | 1.10 | 1.04–1.16 | <0.001 |
| Number of GPs | 0.89 | 0.82–0.97 | <0.01 | 0.96 | 0.90–1.03 | 0.25 |
| Number of consultations | 1.05 | 1.04–1.07 | <0.001 | 1.02 | 1.01–1.03 | <0.001 |
| Number of comorbidities | 1.83 | 1.66–2.01 | <0.001 | 1.48 | 1.38–1.60 | <0.001 |
| Number of years since diagnosis | 0.99 | 0.95–1.03 | 0.66 | 0.96 | 0.93–0.99 | <0.01 |
| | Cholesterol | | | | | |
| | Process indicator | | | Outcome indicator | | |
| Practice type (ref = group practice) | 1.17 | 0.54–2.56 | 0.69 | 1.01 | 0.47–2.17 | 0.98 |
| Practice location (ref = urban) | 0.96 | 0.50–1.84 | 0.90 | 1.24 | 0.65–2.35 | 0.51 |
| Network participation (ref = yes) | 0.59 | 0.25–1.41 | 0.24 | 0.60 | 0.26–1.41 | 0.25 |
| GP gender (ref = male) | 1.14 | 0.75–1.74 | 0.54 | 1.20 | 0.82–1.76 | 0.34 |
| GP age (per 10 years) | 0.81 | 0.66–1.01 | 0.06 | 0.86 | 0.71–1.04 | 0.12 |
| Patient gender (ref = male) | 0.73 | 0.63–0.85 | <0.001 | 0.50 | 0.42–0.58 | <0.001 |
| Patient age (per 10 years) | 0.86 | 0.81–0.91 | <0.001 | 0.92 | 0.86–0.97 | <0.01 |
| Number of GPs | 0.89 | 0.83–0.95 | <0.001 | 0.92 | 0.85–0.98 | <0.05 |
| Number of consultations | 1.00 | 0.99–1.00 | 0.27 | 1.01 | 1.00–1.01 | 0.12 |
| Number of comorbidities | 1.89 | 1.75–2.04 | <0.001 | 1.40 | 1.30–1.51 | <0.001 |
| Number of years since diagnosis | 1.02 | 0.99–1.06 | 0.18 | 1.08 | 1.04–1.12 | <0.001 |

OR = Odds ratio; CI = confidence interval; ref = reference; GP = general practitioner.

33]. However, the finding is contradictory to the guidelines of the American Diabetes Association, recommending a more lenient therapy in multimorbid patients [34]. The same is recommended for disease duration, which we could confirm being associated with lower rates of target value performances for HbA1c. Also notifiable are gender differences regarding QI performance. Female patients had significantly fewer cholesterol measurements and if measured,

## Associations with cumulative quality indicator performance

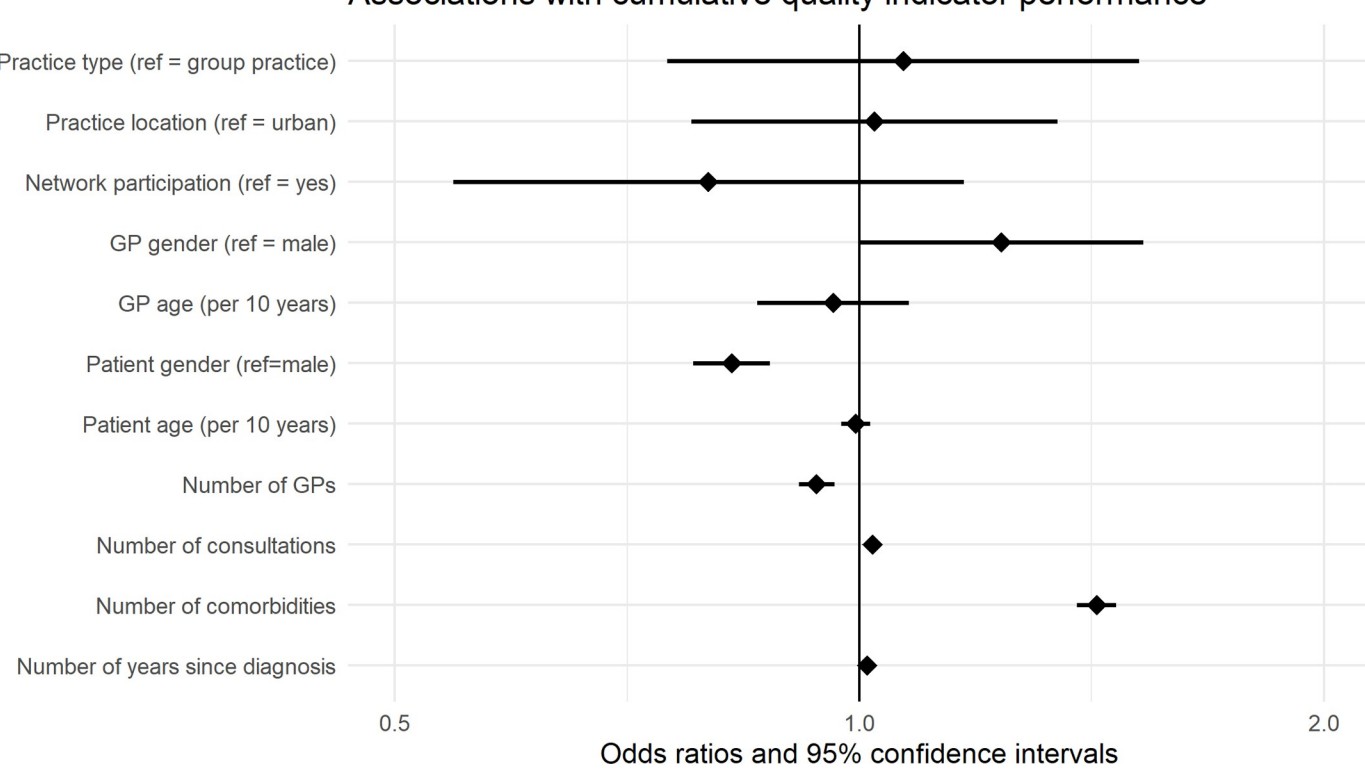

**Fig 2. Associations with cumulative quality indicator performances.** ref = reference; GP = General practitioner.

a higher level. This is highly congruent to recent results investigating gender disparities in diabetes care and revealing gender disparities in risk factor management, prescribing behavior and guideline adherence [35, 36].

Number of consultations was positively associated with QI performance. Van Doorn-Klomberg interpreted similar results in their study as a measure of patients' adherence to a treatment plan, based on an automatic invitation for 3-monthly consultations [13]. In Switzerland, the study of Frei et al. 2013 also found no association with consultation count, which might be due to the fact that in Switzerland a structured invitation mechanism for follow-up appointments is uncommon. Another known factor associated with increased quality of diabetes care is continuity of care [37, 38]. We did not explicitly measure continuity of care, but we found a negative association between number of treating GPs and all three process indicators, and a negative association with the outcome indicator of cholesterol. This finding is not completely in line with Lustman et al. 2016 [37], reporting a positive effect of continuity of care on the major clinical outcomes. Furthermore, we know from literature, that several other patient factors, such as ethnicity/culture, financial resources, beliefs, knowledge and other person-related characteristics, influence diabetes care [12, 39]. Unfortunately, information on these characteristics were not available within the study setting.

In our study, female GPs had a small but positive effect on QI performance, which is repeatedly observed [8, 14]. However, our model also revealed that variation exists between practices and GPs, which cannot be explained by our model. This might be due to a lack of important information about practice and GP characteristics, such as practice culture, working style and accessibility of disease management tools [10, 12, 40]. Overall, it is in line with previous

research, that organization, GP and patient characteristics can only explain small proportions of variation in diabetes care [10, 11].

## Strengths and limitations of this study

A major strength of our study is the relevance of the disease under study: Diabetes is highly prevalent and associated with high morbidity, mortality and costs [41]. Furthermore, we did not allow for exclusion of individual patients and included the entire spectrum of patients with diabetes, reflecting the everyday situation of a GP. We were therefore able to show high generalizability of our data. Another major strength of our study is the wide GP coverage of the FIRE database. FIRE is to date the only database of clinical routine data in Swiss primary care and covers about 10% of GPs practicing in the German-speaking area of Switzerland [42]. GPs participating in the P4P trial are representative for the Swiss GPs in terms of age, but are more often male than the average Swiss GP. However, GPs participating in the FIRE project and additionally in the P4P trial might not be completely representative, as those voluntarily participating in research projects are highly motivated and better performing [43].

This study faces methodological drawbacks commonly present in EMR database studies, i.e. the cross-sectional design, data structure and potential missing data. Not all diabetes-specific comorbidities can be depicted in the FIRE database, due to data structure in the EMR, which does not allow for structural recording of several factors, such as lifestyle, hereditary factors, severity and duration. Duration of diabetes was approximated by using the first record as the onset of diabetes. However, this might highly underestimate the duration, as we have no information from the patient before the GP participated in the FIRE project. Furthermore, information about socioeconomic status and other person- related characteristics are not recorded in the EMR. Missing data is the largest source of bias for our study, as we cannot distinguish between real missing data (not measured) and technical missing data (not available for FIRE due to data capturing or transmission problems for example). This issue is of major concern for the QI measures and comorbidity identification, where we assumed that if there was no positive record, no measurement or disease was present. The proportions not fulfilling the process indicators disclose the maximal percentage of missing data for QI measures. We tried to minimize the amount of missing data by setting a minimum standard of data availability for each GP to be contacted for recruitment. Further limitations of our study are that a few patients were not diagnosed with diabetes over the entire observation period, as the identification needed to be present four months prior to baseline. In addition, type 1 and type 2 patients cannot be fully distinguished due to data structure. However, from the prescription of medications one can conclude that the majority of patients included were patients with type 2 diabetes.

## Conclusion

The influence of practice, GP and patient characteristics on QI performance was surprisingly small and room for improvement in QI performance of Swiss GPs seems to exist in diabetes care. We believe that improving the quality of QI measurements is an important step towards correctly assessing quality of care in primary care. In order to achieve a valid assessment of quality of care, it is essential to comprehensively include all potentially meaningful provider and patient characteristics within routine data collection. Moreover, it will be of particular interest to see whether the P4P approach leads to higher QI performance.

## Supporting information

**S1 Table. Identification scheme for comorbidities and co-medication.** BMI = Body mass index, ICPC = International classification of primary care, ATC = anatomical therapeutic

chemical code, LDL = low density lipoprotein, HDL = High density lipoprotein,
GFR = glomerular filtration rate.
(DOCX)

**S1 Fig. Log OR representing variation from hierarchical multivariable regression analysis for blood pressure process indicator.** GP = general practitioner.
(TIFF)

**S2 Fig. Log OR representing variation from hierarchical multivariable regression analysis for blood pressure outcome indicator.** GP = general practitioner.
(TIFF)

**S3 Fig. Log OR representing variation from hierarchical multivariable regression analysis for HbA1c process indicator.** GP = general practitioner.
(TIFF)

**S4 Fig. Log OR representing variation from hierarchical multivariable regression analysis for HbA1c outcome indicator.** GP = general practitioner.
(TIFF)

**S5 Fig. Log OR representing variation hierarchical multivariable regression analysis for cholesterol process indicator.** GP = general practitioner.
(TIFF)

**S6 Fig. Log OR representing variation from hierarchical multivariable regression analysis for cholesterol outcome indicator.** GP = general practitioner.
(TIFF)

**S7 Fig. Log OR representing variation from hierarchical multivariable regression analysis for cumulative quality indicator.** GP = general practitioner.
(TIFF)

**S1 Checklist.**
(DOC)

## Acknowledgments

We thank the participating GPs for contributing to the present study.

## Author Contributions

**Conceptualization:** Rahel Meier, Oliver Senn, Thomas Rosemann, Corinne Chmiel.

**Data curation:** Fabio Valeri.

**Formal analysis:** Rahel Meier, Fabio Valeri.

**Funding acquisition:** Thomas Rosemann.

**Investigation:** Rahel Meier.

**Project administration:** Rahel Meier.

**Resources:** Thomas Rosemann.

**Software:** Fabio Valeri.

**Supervision:** Oliver Senn, Corinne Chmiel.

**Writing – original draft:** Rahel Meier.

**Writing – review & editing:** Fabio Valeri, Oliver Senn, Thomas Rosemann, Corinne Chmiel.

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
