## [Decision Letter · Decision Letter 0]

6 Jan 2020

PONE-D-19-32658

Quality performance and associated factors in Swiss diabetes care – a cross-sectional study

PLOS ONE

Dear Ms Meier,

Thank you for submitting your manuscript to PLOS ONE. After careful consideration, we feel that it has merit but does not fully meet PLOS ONE’s publication criteria as it currently stands. Therefore, we invite you to submit a revised version of the manuscript that addresses the points raised during the review process.

We would appreciate receiving your revised manuscript by Feb 20 2020 11:59PM. To enhance the reproducibility of your results, we recommend that if applicable you deposit your laboratory protocols in protocols.io, where a protocol can be assigned its own identifier (DOI) such that it can be cited independently in the future. For instructions see: http://journals.plos.org/plosone/s/submission-guidelines#loc-laboratory-protocols

We look forward to receiving your revised manuscript.

Kind regards,

Valérie Pittet, PhD

Academic Editor

PLOS ONE

Journal Requirements:

2. Please ensure you have included the registration number for the clinical trial referenced in the manuscript.

Reviewers' comments:

Reviewer's Responses to Questions

**Comments to the Author**

1. Is the manuscript technically sound, and do the data support the conclusions?

Reviewer #1: Partly

Reviewer #2: Yes

Reviewer #3: Yes

2. Has the statistical analysis been performed appropriately and rigorously? 

Reviewer #1: No

Reviewer #2: Yes

Reviewer #3: Yes

3. Have the authors made all data underlying the findings in their manuscript fully available?

Reviewer #1: No

Reviewer #2: No

Reviewer #3: Yes

4. Is the manuscript presented in an intelligible fashion and written in standard English?

Reviewer #1: Yes

Reviewer #2: Yes

Reviewer #3: Yes

5. Review Comments to the Author

Reviewer #1: Here is a list of specific comments. Note: line and page numbering in reviews and comments is based on ruler applied in Editorial Manager-generated PDF.

1. Page 3, line 68: What was the unit of cluster randomization?

2. Page 3, lines 74–75: What was the difference between the 660000 patients mentioned in this sentence and the 3,833 patients in Figure 1?

3. Page 4, lines 87–88: Please include the definition of “baseline date”.

4. Page 4, lines 96–102: What was the timing of the retrospectively retrieved patients’ data? How was the timing relative to the timing of baseline?

5. Page 5, lines 114–116:

(5a) The use of “population” was not clear. What was the unit (e.g., practice, GP, etc.) that the QI performance was defined?

(5b) The term “indicator” (or more specifically, QI) was not defined previously.

6. Page 5, line 117:

(6a) To this point, the level of practice was not defined clearly. Was the practice selected first and then GP?

(6b) Please define practice type.

7. Page 5, line 118:

(7a) Did “each QI” mean ‘each QI performance’? Please make it clear.

(7b) Did the model include each QI performance as a binary outcome? Please make it clear.

8. Page 5, lines 121–122: Please define the cumulative QI performance clearly.

9. Page 5, line 127: ‘Sample’ would be a better term than “population”.

10. Page 5, line 132: What did “the patient population” refer to?

11. Page 6, Table 1: Please comment on ‘the diabetic status was defined 4 months prior to baseline but QI’s were defined 12 months prior to baseline’. Some patients may not be diabetic when QI’s were collected.

12. Page 7, line 157: Throughout this section, I suggest to write non-significant associations as ‘not find evidence of significant associations’.

13. Page 7, lines 173–174: Please provide more details about variations on practice and on GP level in the Methods section.

14. Page 7, Table 2: Please define the “process” and “outcome” indicators in the Methods section.

Reviewer #2: Please note that I have uploaded my comments as an attachment in order to retain formatting. This should make it easier for the authors to see any edits made to sentences containing typographical errors.

Reviewer #3: 1. The aim of this study was to describe QI performance levels and to analyze associations of QI performance with practice, GP and patient covariates. The authors are to be commended that their research is timely. Diabetes prevalence is rising and the condition increases the risk of morbidity and mortality.

2. It might strengthen the manuscript if the Introduction section were rewritten to link the P4P approach better to the methods and results section and the variables measured and analysed in the current study. It is not clear how financial incentives were taken into account in the Result and Discussion sections.

3. Design and methods. Data regarding participating GPs and the care of their patients are appropriate in relation to the aim of this study. Nevertheless, the GP’s are a rather selected group compared to the total population of GP’s in primary care. The authors may consider to further illuminate potential bias related to selection of participants. As to patient data, there is a lack of information related to clinical variables, ex. diabetes type etc.

4. Results. The data support the conclusion. However, the interpretation of the results should be discussed with more caution. Limitations of the study are not sufficiently outlined.

5. The discussion section is interesting. However, a wider perspective in the discussion section might benefit the manuscript. If the literature references could include more studies from other countries in addition to Swiss primary care, it might be a more interesting paper also for international readers.

6.In conclusion, this is an interesting and timely manuscript. More effort might however, be put in further clarification about the sample population, and how selection bias might have affected the results.

6. PLOS authors have the option to publish the peer review history of their article (what does this mean?). If published, this will include your full peer review and any attached files.

Reviewer #1: No

Reviewer #2: No

Reviewer #3: No

---

## [Author Response · Author response to Decision Letter 0]

28 Feb 2020

Dear Dr. Pittet,

Thank you for considering our manuscript.

We were very glad to receive the reviewers’ thoughtful and constructive comments. These allowed us to substantially improve the manuscript and to further elaborate its content. Please find the point-by-point response table attached as well as a tracked-changes version and a clean version of the revised manuscript.

Further, we addressed the journal requirements on style requirements, inclusion of registration number for the clinical trial and data availability statement. We updated our data availability statement as follows: 

Legal restrictions in Switzerland prohibit public release of original patient data without consent. The authors’ fully anonymized data is exempt from these legal restrictions. However, the data could be deanonymized by individuals or organizations, such as health insurers, which have overlapping data (e.g. patient year of birth and consultation dates). Data access queries can be addressed to Rahel Meier (rahel.meier@usz.ch) after clearance by the local ethics committee or to the Kantonale Ethikkommission Zurich (Local Ethics Committee of the Canton of Zurich) (Info.KEK@kek.zh.ch).

Unfortunately, we are not allowed to make our data publicly available, even after removing certain types of information. With the addition of further publications, the entire database would gradually become accessible.

We hope our revision to be satisfactory and sufficient for the manuscript to be accepted for publication in Plos One.

We look forward to receiving your response. 

Kind regards on behalf of all authors,

Rahel Meier

---

## [Decision Letter · Decision Letter 1]

27 Mar 2020

PONE-D-19-32658R1

Quality performance and associated factors in Swiss diabetes care – a cross-sectional study

PLOS ONE

Dear Ms Meier,

Thank you for submitting your manuscript to PLOS ONE. After careful consideration, we feel that it has merit but does not fully meet PLOS ONE’s publication criteria as it currently stands. Therefore, we invite you to submit a revised version of the manuscript that addresses the points raised during the review process.

We would appreciate receiving your revised manuscript by May 11 2020 11:59PM. To enhance the reproducibility of your results, we recommend that if applicable you deposit your laboratory protocols in protocols.io, where a protocol can be assigned its own identifier (DOI) such that it can be cited independently in the future. For instructions see: http://journals.plos.org/plosone/s/submission-guidelines#loc-laboratory-protocols

We look forward to receiving your revised manuscript.

Kind regards,

Valérie Pittet, PhD

Academic Editor

PLOS ONE

**Comments to the Author**

1. If the authors have adequately addressed your comments raised in a previous round of review and you feel that this manuscript is now acceptable for publication, you may indicate that here to bypass the “Comments to the Author” section, enter your conflict of interest statement in the “Confidential to Editor” section, and submit your "Accept" recommendation.

Reviewer #1: All comments have been addressed

Reviewer #2: (No Response)

Reviewer #3: All comments have been addressed

2. Is the manuscript technically sound, and do the data support the conclusions?

Reviewer #1: (No Response)

Reviewer #2: Yes

Reviewer #3: Yes

3. Has the statistical analysis been performed appropriately and rigorously? 

Reviewer #1: Yes

Reviewer #2: Yes

Reviewer #3: Yes

4. Have the authors made all data underlying the findings in their manuscript fully available?

Reviewer #1: No

Reviewer #2: No

Reviewer #3: Yes

5. Is the manuscript presented in an intelligible fashion and written in standard English?

Reviewer #1: Yes

Reviewer #2: Yes

Reviewer #3: Yes

6. Review Comments to the Author

Reviewer #1: Here is a list of specific comments. Note: line and page numbering in reviews and comments is based on ruler applied in Editorial Manager-generated PDF.

1. Page 7, lines 134–136: The cumulative QI performance ranged from 0 to 6. Please clarify the use of the hierarchical multivariable “binomial” regression model for an ordinal outcome. Did you mean a hierarchical multivariable ordinal logistic regression model where the odds would be interpreted as the odds of having more QI performance?

Reviewer #2: Thank you for making these changes. I only have some small further suggestions regarding the manuscript. As an aside, please note when responding to reviewer comments in future it is very helpful to firstly, provide some detail on the nature of the revision made and secondly, to indicate the line numbers where the revisions have been made. This makes it much easier (and quicker) for the reviewer to assess the changes.

In my previous review I suggested clarifying what you mean by “Structural data on participating GP are collected at individual project entry.” (line 75). Thank you for adding further detail to explain that this was done manually. However, I feel you should tell the reader who collected this information - I presume the research team?

There are still some typos in places throughout the manuscript so would advise careful proofreading before submission e.g. However, one can indirectly conclude from medication prescriptions that, the majority of included patients were patients with type 2 diabetes.

In the previous review I suggested rephrasing this sentence (line 188) as it as unclear what was meant. I suggest further rewording as this could be clearer and have included amendment:

“Number of years since diabetes diagnosis was much shorter in our study, which is explained by the fact limited data were available from before GPs participated in the FIRE project.”

In my previous review I noted that the Stone et al study (line 195) was missing a reference and that I thought this sentence was meant to be connected to the previous line. Thank you for adding the reference to the study. It is always helpful for the reader to have the reference cited a that place in the text even if the article was introduced earlier in the manuscript, particularly as you are using Vancouver referencing style. You did not address the second part of this comment. The sentence begins with ‘whereas’ and so hence my question about whether it should be connected to the preceding sentence. For example, I suggest it would read as:

"The proportions of patients fulfilling the process and outcome indicators for BP and HbA1c were highly comparable to the methodological similar study of van Doorn-Klomberg et al. 2015 [13], whereas the European cross-sectional study by Stone et al. 2013 [27] reported process indicators above 90%."

Thank you for amending line 224. However, this sentence is still unclear. I think it should read as follows such that you clarify the direction of the effect:

"This finding is not completely in line with Lustman et al. 2016 [37], reporting a positive effect of continuity of care on the major clinical outcomes"

Reviewer #3: The authors have adequately addressed all points I raised. The revised manuscript has improved, is well written and an interesting piece of research. I have no further comments.

7. PLOS authors have the option to publish the peer review history of their article (what does this mean?). If published, this will include your full peer review and any attached files.

Reviewer #1: No

Reviewer #2: No

Reviewer #3: No

---

## [Author Response · Author response to Decision Letter 1]

3 Apr 2020

1.1 Page 7, lines 134–136: The cumulative QI performance ranged from 0 to 6. Please clarify the use of the hierarchical multivariable “binomial” regression model for an ordinal outcome. Did you mean a hierarchical multivariable ordinal logistic regression model where the odds would be interpreted as the odds of having more QI performance? 

Thank you for your comment. By binomial regression, we meant that the dependent variable was defined as the – binomially distributed - cumulative QI performance: (cumulative QI performance, 6- cumulative QI performance), where the odds would be interpreted as the odds of satisfying an individual QI. We added more details to the manuscript (lines 139 – 140 in the manuscript with tracked changes) and hope that it is now better understandable what we meant. 

2.1 In my previous review I suggested clarifying what you mean by “Structural data on participating GP are collected at individual project entry.” (line 75). Thank you for adding further detail to explain that this was done manually. However, I feel you should tell the reader who collected this information - I presume the research team? 

Thank you for your comment. We clarified further, as you suggested, that the GPs self-reported the structural information to our study nurse (lines 78 - 80 in the manuscript with tracked changes) 

2.2 There are still some typos in places throughout the manuscript so would advise careful proofreading before submission e.g. However, one can indirectly conclude from medication prescriptions that, the majority of included patients were patients with type 2 diabetes. 

Thank you for pointing out this specific sentence. We made corrections to this sentence and carefully proofread the whole manuscript and made some minor changes throughout. 

2.3 In the previous review I suggested rephrasing this sentence (line 188) as it as unclear what was meant. I suggest further rewording as this could be clearer and have included amendment:

“Number of years since diabetes diagnosis was much shorter in our study, which is explained by the fact limited data were available from before GPs participated in the FIRE project.” 

Thank you for your suggestion and for clarifying its meaning. We changed the sentence as you suggested (lines 217 -219 in the manuscript with tracked changes).

2.4 In my previous review I noted that the Stone et al study (line 195) was missing a reference and that I thought this sentence was meant to be connected to the previous line. Thank you for adding the reference to the study. It is always helpful for the reader to have the reference cited a that place in the text even if the article was introduced earlier in the manuscript, particularly as you are using Vancouver referencing style. You did not address the second part of this comment. The sentence begins with ‘whereas’ and so hence my question about whether it should be connected to the preceding sentence. For example, I suggest it would read as:

"The proportions of patients fulfilling the process and outcome indicators for BP and HbA1c were highly comparable to the methodological similar study of van Doorn-Klomberg et al. 2015 [13], whereas the European cross-sectional study by Stone et al. 2013 [27] reported process indicators above 90%." 

Thank you for your suggestion. We adapted the sentence as you suggested (line 224 in the manuscript with tracked changes).

2.5 Thank you for amending line 224. However, this sentence is still unclear. I think it should read as follows such that you clarify the direction of the effect:

"This finding is not completely in line with Lustman et al. 2016 [37], reporting a positive effect of continuity of care on the major clinical outcomes 

Thank you for your comment. We changed the sentence as you suggested and clarified the direction of the effect (line 253 in the manuscript with tracked changes).

---

## [Editor Report · Decision Letter 2]

21 Apr 2020

Quality performance and associated factors in Swiss diabetes care – a cross-sectional study

PONE-D-19-32658R2

Dear Dr. Meier,

We are pleased to inform you that your manuscript has been judged scientifically suitable for publication and will be formally accepted for publication once it complies with all outstanding technical requirements.

With kind regards,

Valérie Pittet, PhD

Academic Editor

PLOS ONE
---

## [Editor Report · Acceptance letter]

24 Apr 2020

PONE-D-19-32658R2 

Quality performance and associated factors in Swiss diabetes care – a cross-sectional study 

Dear Dr. Meier:

I am pleased to inform you that your manuscript has been deemed suitable for publication in PLOS ONE. Congratulations! Your manuscript is now with our production department. 

With kind regards,

on behalf of

PD Dr. Valérie Pittet 

Academic Editor

PLOS ONE